# Cholesterol Metabolism and Urinary System Tumors

**DOI:** 10.3390/biomedicines12081832

**Published:** 2024-08-12

**Authors:** Songyuan Yang, Zehua Ye, Jinzhuo Ning, Peihan Wang, Xiangjun Zhou, Wei Li, Fan Cheng

**Affiliations:** 1Department of Urology, Renmin Hospital of Wuhan University, Wuhan 430060, China; ysy097019@163.com (S.Y.); yezehua0704@163.com (Z.Y.); njz120511@whu.edu.cn (J.N.); 18327129949@139.com (P.W.); zxj19840902@163.com (X.Z.); 2Department of Anesthesiology, Renmin Hospital of Wuhan University, Wuhan 430060, China; rm000834@whu.edu.cn

**Keywords:** cholesterol metabolism, urinary system tumors, renal cancer, bladder cancer, prostate cancer

## Abstract

Cancers of the urinary system account for 13.1% of new cancer cases and 7.9% of cancer-related deaths. Of them, renal cancer, bladder cancer, and prostate cancer are most prevalent and pose a substantial threat to human health and the quality of life. Prostate cancer is the most common malignant tumor in the male urinary system. It is the second most common type of malignant tumor in men, with lung cancer surpassing its incidence and mortality. Bladder cancer has one of the highest incidences and is sex-related, with men reporting a significantly higher incidence than women. Tumor development in the urinary system is associated with factors, such as smoking, obesity, high blood pressure, diet, occupational exposure, and genetics. The treatment strategies primarily involve surgery, radiation therapy, and chemotherapy. Cholesterol metabolism is a crucial physiological process associated with developing and progressing urinary system tumors. High cholesterol levels are closely associated with tumor occurrence, invasion, and metastasis. This warrants thoroughly investigating the role of cholesterol metabolism in urinary system tumors and identifying novel treatment methods for the prevention, early diagnosis, targeted treatment, and drug resistance of urinary system tumors.

## 1. Introduction

Cancers of the urinary system comprise 13.1% of all new cancer diagnoses and are responsible for 7.9% of cancer-related fatalities. Of these, the most prevalent are renal cancer (KC), bladder cancer (UBC), and prostate cancer (PCa). With the expansion and aging of the global population, these cancers are becoming frequent. Compared with 1990, the global incidence of KC, UBC, and PCa has increased by 155%, 123%, and 169%, respectively [1,2]. Urinary system tumors pose a substantial threat to human health and quality of life. In particular, prostate cancer is the most common malignant tumor in the male urinary system, with incidence and mortality second only to lung cancer in men. Bladder cancer has one of the highest incidences and is sex-related, with men reporting a significantly higher incidence than women [3]. Urinary system tumors have been associated with factors, such as smoking, obesity, hypertension, diet, occupational exposure, and genetics [2,4,5,6]. The treatment options primarily include surgery, radiotherapy, and chemotherapy [7,8]. However, their effectiveness and survival rates are low. Numerous patients experience tumor recurrence and drug resistance, and the adverse effects of the treatments significantly impact their quality of life. Therefore, researchers should explore novel mechanisms underlying the development and progression of urinary system tumors, providing novel insights and potential approaches for their treatment [9].

Cholesterol metabolism is a crucial physiological process associated with developing and progressing urinary system tumors. High cholesterol levels have been strongly associated with tumor occurrence, invasion, and metastasis [10].

Cholesterol, a derivative of cyclopentane polyhydrophenanthrene, is extensive in the human body. It is embedded in the cell membrane, providing cell rigidity. Additionally, cholesterol determines cell permeability to ions: the higher the cholesterol on the membrane, the lower its permeability. Therefore, membrane cholesterol concentration can reflect the function of the membrane. Mitochondria with high permeability have low cholesterol concentrations, whereas those with limited permeability have higher concentrations. Cholesterol enables numerous cell receptors and transporters to randomly drift on the cell membrane. Moreover, it can form “rafts”, organizing signal molecules and regulating membrane protein transport and neural transmission [11]. Additionally, cholesterol is the precursor to steroid hormones, including glucocorticoids and mineralocorticoids, sex hormones, and vitamin D. These hormones regulate the balance of carbohydrates, sodium, reproduction, and skeletal health. Additionally, cholesterol is the precursor to bile acids, crucial for the intestinal absorption of dietary fats and energy and glucose metabolism regulation [12]. Malignant tumor cells can proliferate rapidly. Cancer cells require high cholesterol to meet the demands of membrane biogenesis and other functional needs. For example, patients with breast cancer report accumulated tumor metabolite 6-oxo-cholesterol-3β,5α-diol, which is derived from cholesterol. This metabolite binds to glucocorticoid receptors, subsequently promoting tumor growth [13]. For example, cholesterol on the cell membrane regulates the proliferation and migration of tumor cells and their response to chemotherapy drugs. Cholesterol in the mitochondria activates the mammalian target of rapamycin complex 1 axis, thereby promoting tumor cell survival. Additionally, cholesterol and its metabolites impact the tumor microenvironment [10,14]. In summary, cholesterol metabolism significantly promotes cancer progression, including cell proliferation, migration, and invasion [15,16]. Cholesterol consumption or transport obstruction can hinder tumor growth and invasion in various types of cancer [17,18]. This article provides an overview of the mechanisms underlying cholesterol homeostasis and the carcinogenic mechanisms underlying abnormal cholesterol metabolism in the three major urinary system tumors as well as corresponding anti-tumor treatments.

## 2. Normal Cholesterol Metabolism

Cholesterol is essential to maintaining cellular homeostasis. It serves as a precursor to steroid hormones and is a crucial component of the plasma membrane. Additionally, it is abundant in lipid rafts and central to intracellular signal transduction [19].

The biosynthesis pathway of cholesterol involves a complex biochemical process requiring over 15 enzymes [20,21]. This pathway can be divided into two stages as follows: (1) isoprene unit condensation to form a 30-carbon molecule, squalene; (2) squalene cyclization to form lanosterol, which is eventually converted into cholesterol [20,21,22,23]. First, this pathway involves one molecule each of acetyl coenzyme A (2C) and acetoacetyl coenzyme A, which are catalyzed by 3-hydroxy-3-methylglutaryl-coenzyme A synthase to form a six-carbon 3-hydroxy-3-methylglutaryl-coenzyme A (HMG-CoA). Second, the membrane-bound enzyme 3-hydroxy-3-methylglutaryl-coenzyme A reductase (HMGCR) converts HMG-CoA into mevalonic acid. This reaction is the primary rate-controlling step in cholesterol biosynthesis and a major target of statin drugs. Cholesterol biosynthesis is regulated by a feedback inhibition system. In this system, the cholesterol levels within the cell are detected, leading to the regulated expression of key proteins that control cholesterol homeostasis [24]. Sterol regulatory element-binding protein (SREBP), primarily type 2 (SREBP2) is the most crucial component. For high cellular cholesterol levels, SREBP2 binds to the endoplasmic reticulum membrane anchor protein Insig-1 in the form of a complex with SREBP2 cleavage-activating protein (SCAP). It is located in the endoplasmic reticulum. However, for depleted sterols, SCAP facilitates SREBP2 transfer from the endoplasmic reticulum to the Golgi apparatus (Figure 1). Simultaneously, the complex is cleaved, releasing a portion of the protein from the membrane. Activated SREBP2 enters the nucleus and activates the transcription of multiple genes regulating cholesterol synthesis by binding to the sterol response elements in the enhancer/promoter region. Blood cholesterol is primarily transported by low-density lipoprotein (LDL), and the LDL receptor (LDLR) on the cell membrane mediates its endocytosis into the cell. Excess cholesterol within the cell can be reduced by inhibiting LDLR replenishment, thereby reducing cholesterol uptake from the blood. Excess cholesterol can be converted into cholesterol esters for storage or transformed into steroid hormones. Moreover, it can be transported into the blood via the ATP-binding cassette transporter A1 (ABCA1) [25,26].

Cholesterol synthesis, uptake, composition, and transformation in human tissues are precisely regulated. Disruption can lead to an imbalance in cholesterol metabolism, potentially leading to disease onset (Figure 2).

## 3. Cholesterol Metabolism and Urinary System Tumors

### 3.1. Bladder Cancer

UBC is the 10th most commonly diagnosed cancer worldwide [27]. Its incidence is steadily increasing globally [28]. Smoking is the most significant risk factor, accounting for approximately half of all cases. Other risk factors include occupational exposure to aromatic amines, polycyclic aromatic hydrocarbons, and chlorinated hydrocarbons as well as arsenic exposure and the chlorination of drinking water [27]. Approximately 90% of bladder cancer cases are urothelial carcinomas. Thus, the pathological subtype in UBC is urothelial carcinoma, unless otherwise specified [29]. Additionally, UBC is categorized into non-muscle invasive bladder cancer (NMIBC), muscle-invasive bladder cancer, and the advanced stages of late-stage bladder cancer [27]. NMIBC treatment involves surgery combined with chemotherapy. The treatment of bladder cancer has significantly improved; nonetheless, approximately 60% to 70% of patients with NMIBC experience a relapse within 3 years of tumor resection [30,31]. This warrants novel treatment strategies for bladder cancer. Several researchers have explored the association of cholesterol metabolism with bladder cancer occurrence, development, treatment, and prognosis. Increased cholesterol intake has been associated with an increased risk of bladder cancer, and total cholesterol levels have been positively correlated with total cancer mortality. Moreover, cholesterol metabolism has been associated with the induction of cancer cell proliferation and distant metastasis [32,33,34,35,36,37].

The pathogenesis of bladder cancer is unclear; however, factors, such as smoking, obesity, environmental factors, and lifestyle, are closely associated with its onset. Cholesterol and its metabolism may be one of the potential mechanisms. The overexpression of 7-dehydrocholesterol reductase in cholesterol metabolism can accelerate tumor growth in the G0/G1 phase through the phosphoinositide 3 kinase/protein kinase B (AKT)/mTOR signaling pathway, can antagonize cell apoptosis, and can enhance cell invasion and migration capabilities and epithelial–mesenchymal transition [38]. 25-hydroxycholesterol, a cholesterol oxidation product, is elevated in bladder cancer tissues; it promotes the proliferation and epithelial–mesenchymal transformation of human T24 and RT4 bladder cancer cells. 25-hydroxycholesterol enhances the resistance of T24 and RT4 cells to doxorubicin (experimental research) [39]. The nuclear transcription factor Y subunit γ (NFC)-37 interacts with cyclic AMP response element-binding protein and SREBP2, activates the mevalonic acid pathway transcription, promotes cholesterol biosynthesis, and promotes cell proliferation and tumor growth. Therefore, statin drugs targeting the mevalonic acid pathway can inhibit NFYC-37-induced cell proliferation and tumor growth (experimental research) [40]. The overexpression of farnesoid X receptor (FXR) activates 5′ amp-activated protein kinase (AMPK) and downregulates SREBP2 and HMGCR expression to reduce cholesterol biosynthesis and secretion. Additionally, dorsomorphin, an AMPK inhibitor, can reverse the inhibitory effect of FXR overexpression on migration, invasion, and angiogenesis. The treatment of FXR overexpression combined with atorvastatin enhances the downregulation of the migration, adhesion, invasion, and angiogenesis characteristics of human urothelial carcinoma (experimental research) [41]. ox-LDL can associate high cholesterol blood disease with UBC progression by enhancing cancer stem cells. Ezetimibe can reverse diet-induced high cholesterol blood disease and carcinogenesis by inhibiting intestinal cholesterol absorption. Therefore, reducing serum ox-LDL or targeting the cluster of differentiation 36/Janus kinase 2/signal transducer and activator of transcription 3 axis may be a potential treatment strategy for UBC with high cholesterol blood disease (experimental research) [42]. However, the incidence and mortality of malignant tumors in patients receiving ezetimibe treatment are similar to those in controls (clinical research) [43]. The levels of cholesterol and its metabolic products can be used to predict bladder cancer prognosis [39,44].

However, the risk of UBC does not change between statin users and non-users. Moreover, the use of statins is not associated with local control, recurrence, survival, or mortality in patients with UBC. Statin use does not affect the effectiveness of Bacillus Calmette–Guérin immunotherapy [45]. Their long-term use can increase the risk of bladder cancer [46].

### 3.2. Prostate Cancer

PCa is a common malignant tumor and the second leading cause of cancer-related deaths in men [47]. The correlation between PCa and obesity-related chronic inflammatory state is becoming increasingly clear [48]. Importantly, the incidence of PCa will significantly increase in the next decades, posing substantial health concerns. This can be attributed to the gradual aging of the population, the high prevalence of obesity, and widespread Western dietary habits in developed countries. Approximately 80% to 90% of patients with PCa are diagnosed as local or locally advanced, most of which can be treated locally with surgery or radiotherapy, with or without androgen deprivation therapy [49]. However, approximately 10% of the patients have metastases during initial diagnosis. These patients are referred to as patients with newly diagnosed metastatic castration-sensitive PCa (mCSPC). Their overall survival is shorter than patients who developed metastases a few years after the initial diagnosis [50,51]. Importantly, prostate-specific antigen testing is used as a screening method; however, the number of patients with mCSPC is expected to increase. In the past decade, metastatic castration-resistant prostate cancer (mCRPC) treatment has undergone tremendous changes with the introduction of novel treatment strategies, such as immunotherapy, poly (ADP-ribose) polymerase (PARP) inhibitor (PARPi), androgen receptor signaling inhibitor (ARSI), Abiraterone (a specific cytochrome 17α-hydroxylase inhibitor that is critical for androgen synthesis), and Enzalutamide (a potent nonsteroidal AR inhibitor that binds to AR with high affinity). Nonetheless, mCRPC prognosis remains poor, primarily because of its heterogeneity and resistance to treatment. Therefore, researchers should determine predictive biomarkers to guide the optimal sequence of systemic treatment and identify novel therapeutic targets. Cholesterol levels are associated with the occurrence and development of prostate cancer, and this association has racial differences (total cholesterol was associated with higher fatal prostate cancer risk in White men only. ApoA was associated with higher fatal prostate cancer risk overall, but ApoB was associated with higher fatal prostate cancer risk in Black men only). Cholesterol and its metabolic products have become the potential therapeutic targets for prostate cancer [52,53,54,55,56].

The control of cholesterol imbalance is associated with invasive or late-stage prostate cancer. Multiple mechanisms underlie cholesterol accumulation and how it promotes prostate cancer development, including changes in cholesterol metabolism enzymes, particularly HMG-CoA reductase, enhanced blood uptake, impaired LDLR regulation, and cholesterol imbalance mediated through ABCA1 [57,58,59,60,61,62].

Cholesterol and its metabolic products are being investigated to identify the therapeutic targets for prostate cancer. Kalogirou et al. (experimental research) [63] demonstrated that squalene epoxidase (SQLE), an enzyme in the cholesterol biosynthesis pathway, is overexpressed in late-stage PCa. SQLE expression is controlled by microRNA 205 (miR-205), which is significantly downregulated in late-stage PCa. Restoring miR-205 expression or competitively inhibiting SQLE can inhibit de novo cholesterol biosynthesis. The Food and Drug Administration-approved antifungal drug terbinafine effectively blocks the growth of in situ tumors in mice by inhibiting SQLE. Therefore, SQLE can be used as a therapeutic target. Hryniewicz-Jankowska et al. (experimental research) [64] reported that cholesterol is central to the formation of membrane rafts in prostate cancer development; thus, it can be used as a target for cancer treatment and prevention. Plant-derived polyphenols are crucial in prostate cancer prevention and treatment. Zhou et al. (experimental research) [65] demonstrated that acyl-coenzyme A synthetase short-chain family member 3 inhibits prostate cancer by downregulating lipid droplet-associated protein Perilipin 3. Gan et al. [66,67,68] reported that proprotein convertase subtilisin/kexin type 9 regulates radiosensitivity through the mitochondrial pathway, highlighting it as a therapeutic target for prostate cancer. Wang, X. et al. (experimental research) [69] demonstrated that excessive cholesterol accumulation is associated with PCa. In a high-grade prostate intraepithelial neoplasia mouse model, the overexpression of fatty acid synthase and ABCA1 knockout led to prostate intraepithelial neoplasia progressing to invasive PCa with a 100% penetrance rate. Simultaneously, the number of prostate cancer stem cells increased, accompanied by the activation of the prostaglandin E2 and transforming growth factor-β signaling pathways. The steady increase in PCa incidence and mortality in the Chinese population may be attributed to the combined effects of fatty acids and cholesterol. Reducing the intake of dietary fat and cholesterol can decelerate the progression of latent lesions to PCa (experimental research) [70]. Fatostatin inhibits the separation of SCAP/SREBP from INSIGN by binding it to SCAP. This step blocks the activation of SREBP1 and SREBP2, inhibits cell proliferation and induces apoptotic death [71,72,73]. Two months of fatostatin treatment can inhibit PCa proliferation and distant lymph node metastasis (experimental research) [74]. Betulin and xanthohumol are natural SCAP/SREBP translocation inhibitors. Betulin induces the interaction of SCAP-INSIGN, and xanthohumol blocks the binding of SREBP in coat protein complex II vesicles, which are possible approaches to treating PCa [75,76]. Similar to other cancers, statins offer benefits for patients with PCa [77,78,79,80,81,82]. In summary, PCa treatment by regulating cholesterol and its metabolic products is receiving more research attention and may become a novel treatment approach.

### 3.3. Kidney Cancer

KC is the 14th most common malignant tumor worldwide. KC is the ninth most common cancer in men and the 14th most common cancer in women [83]. Renal cell carcinoma (RCC) accounts for the majority (90%) of KC cases, primarily including clear cell RCC (ccRCC; 70%), papillary RCC (10–15%), and chromophobe RCC (5%) [84]. ccRCC accounts for the majority of cases; it is characterized by the accumulation of cholesterol, cholesterol esters, other neutral lipids, and glycogen. KC is associated with obesity, hypertension, smoking, and genetics [85]. KC treatment primarily depends on surgery and drugs; however, the effect of drugs is limited. The ability of kidney tumors to maintain cholesterol homeostasis may be a key factor in the drug resistance of patients with KC to TKIs (experimental research) [86]. ccRCC may overexpress lysosomal acid lipase, activate the Akt/Src pathway, increase the level of 14,15-epoxyeicosatrienoic acid, and promote cell proliferation and survival (experimental research) [87]. Cholesterol promotes renal cancer cell migration and invasion by regulating the Krüppel-like factor 5/miR-27a/FBXW7 pathway (experimental research) [88]. Additionally, serum cholesterol levels can reflect the prognosis of patients with ccRCC [89,90,91].

Blood cholesterol levels have been associated with KC, particularly the clear cell carcinoma subtype (clinical research) [92]. ccRCC growth and survival depend on exogenous cholesterol. Additionally, an increase in circulating high-density lipoprotein (HDL) cholesterol increases the risk of ccRCC. An increase in dietary cholesterol intake promotes tumor growth. These observations have implications for targeting cholesterol transporters and managing the circulating cholesterol in patients with ccRCC. Gene sets associated with cholesterol metabolism and biosynthesis are significantly downregulated in ccRCC, indicating that ccRCC tumor cells depend on the cholesterol input. A genome-wide association study conducted a two-sample Mendelian randomization in 10,784 patients with RCC and 20,406 without RCC; the probability of RCC significantly increased when gene alleles predicted an increase in serum HDL or HDL cholesterol levels. In vivo, mice injected with A498 cells and fed a high-cholesterol diet demonstrate larger tumor volumes than those fed a cholesterol-free diet. The cholesterol input of human clear cell carcinoma cells primarily depends on the transport protein scavenger receptor B1. Moreover, acyl-coenzyme A:cholesterol acyltransferase 1 (ACAT-1) is strongly expressed in ccRCC. ACAT-1 upregulation increases ACAT enzyme activity, thereby accelerating the accumulation of cholesterol esters in ccRCC. This phenomenon can be used to develop novel drug targets in the future [93,94,95,96]. Furthermore, Cytochrome P450 Family 27 Subfamily A Member 1 exerts a tumor-suppressive effect on RCC, which can be used to develop treatment approaches (experimental research) [97].

## 4. Targeting Cholesterol Metabolism for Cancer Therapy

Impeding active cholesterol metabolism, for example, by inhibiting the mevalonate pathway, is a feasible anti-tumor strategy (Table 1). This can be attributed to the functions of cholesterol metabolism in cancer progression. The key therapeutic targets are as follows: targeting cholesterol synthesis, cholesterol esterification, liver X receptor (LXR) signaling, and combination strategies. For example, Larsen et al. (clinical research) [98] demonstrated that statin use is associated with reduced PCa-related mortality. Lee and co-workers reported that inhibited cholesterol esterification suppresses PCa metastasis by impairing the wingless-related integration Wnt/β-catenin pathway. Thus, they used Avasimibe to target ACAT1 and suppress tumors (experimental research) [99,100]. Furthermore, Lee and co-workers found that Avasimin can inhibit PC3 prostate cancer by Increasing cell apoptosis with no clear cytotoxicity (experimental research) [101]. Wu and co-workers targeted the transcription factor receptor LXR to treat ccRCC (experimental research) [102]. Enzalutamide combined with simvastatin can inhibit PCa by restricting surface receptor signaling, thus sensitizing the blockade of human epidermal growth factor receptor 2 or androgen receptors [103] (Figure 3). While several clinical and basic studies have found that drugs targeting cholesterol metabolism can inhibit the progression of genitourinary tumors, evidence remains limited. Future research is needed for further validation, as well as the exploration of other drugs targeting cholesterol metabolism.

## 5. Conclusions and Future Directions

Cholesterol and its metabolic products are associated with urinary system tumors. Lowering cholesterol levels or blocking cholesterol metabolism inhibits tumor growth. Statins have been used in anti-tumor treatment; however, related drugs pose problems, such as uncertain efficacy, low specificity, and high toxicity. Additionally, a decrease in serum cholesterol may be associated with poor ccRCC prognosis. Cholesterol is necessary for metabolism; therefore, cholesterol levels should not be blindly reduced. With the development and progress of modern biomedical technology and the integration of multidisciplinary achievements, emerging treatment methods, such as molecularly targeted drugs, immunotherapy, and combined chemotherapy have numerous application prospects. Cholesterol and its metabolic products have unique advantages in adjuvant medication and targeting drug-resistant tumor cells and could provide a new avenue for the treatment of drug-resistant urological tumors. Current treatment strategies, such as targeting cholesterol synthesis, cholesterol esterification, liver X receptor (LXR) signaling, and combination strategies, have shown great promise for treating urinary system tumors. However, in the future, interventions targeting other cholesterol metabolism points can still be explored. Cholesterol and its metabolic products will be central to the treatment of urinary system tumors in the future.

## Figures and Tables

**Figure 1 biomedicines-12-01832-f001:**
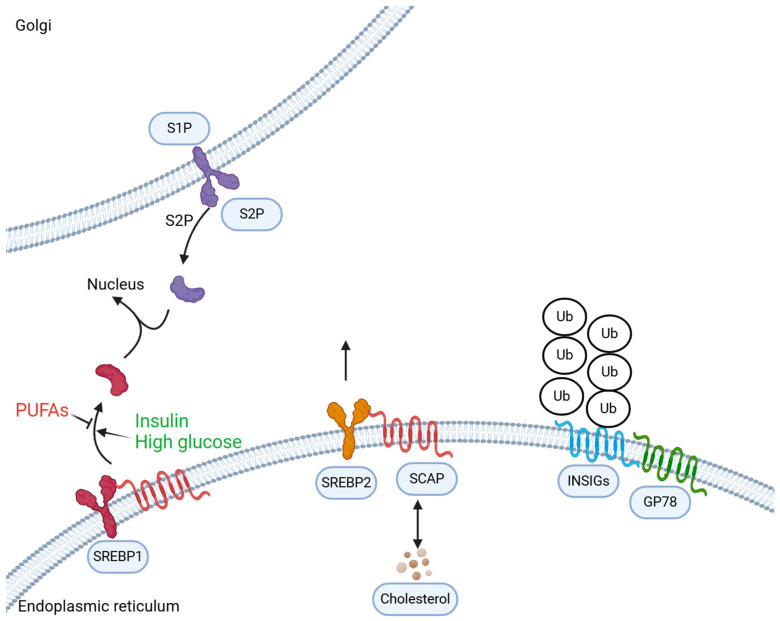
The sterol regulatory element-binding protein pathway. In the presence of cholesterol and oxycholesterols (25-hydroxycholesterol and 27-hydroxycholesterol), the sterol regulatory element-binding protein 2 (SREBP2)–SREBP cleavage-activating protein (SCAP) complex is retained in the endoplasmic reticulum together with insulin-induced gene proteins (INSIGs). In the absence of sterols, INSIGs become ubiquitylated (Ub) and are rapidly degraded. After transport to the Golgi, two proteolytic cleavage enzymes, site-1 protease (S1P) and S2P, release the N-terminal domain of SREBP2. The proteolysis of SREBP1 is not strongly sterol-regulated but rather is inhibited by polyunsaturated fatty acids (PUFAs) and induced by insulin or high-glucose conditions. SREBP1 activation remains incompletely understood. GP78, E3 ubiquitin-protein ligase AMFR.

**Figure 2 biomedicines-12-01832-f002:**
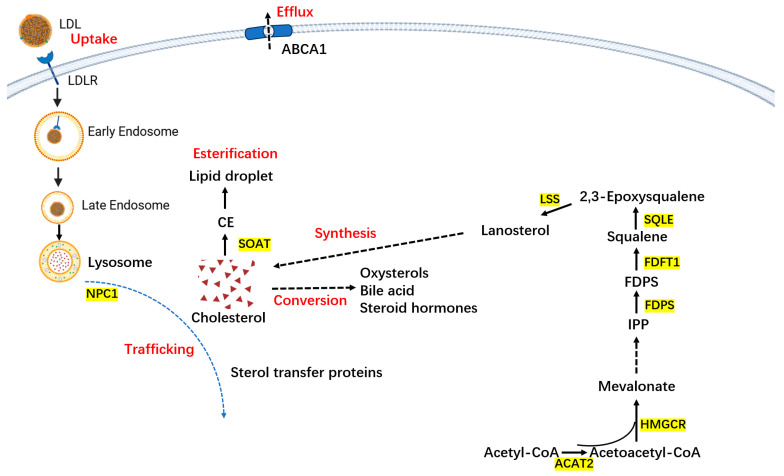
The homeostasis of intracellular cholesterol metabolism. The major pathways of cholesterol metabolism in most normal cells and cancer cells include cholesterol biosynthesis, uptake, efflux, conversion, and esterification. Intracellular cholesterol content is precisely controlled by those pathways as well as cholesterol trafficking. (a) The biosynthesis pathway converts acetyl-CoA into cholesterol through nearly 30 enzymatic reactions, among which HMGCR and squalene epoxidase (SQLE) are the two key speed-limiting enzymes. ACAT2: Acetyl Coenzyme A Acetyltransferase 2, FDFT1: Farnesyl-diphosphate farnesyltransferase 1. (b) Besides de novo biosynthesis, most cells absorb cholesterol from low-density lipoprotein (LDL) via LDLR-mediated endocytosis. (c) Cholesterol derived from LDL is then transferred to the endosomes and lysosomes and is finally delivered to plasma membrane and endoplasmic reticulum with the help of Niemann–Pick disease type C1/2(NPC2/NPC1) and sterol transfer proteins. (d) Elevated cholesterol can be converted to cholesteryl ester (CE) by sterol O-acyltransferase (SOAT) and stored in lipid droplets or catalyzed to produce oxysterols, bile acids, and steroid hormones. (e) Excess cholesterol in cells is excreted to extracellular via ATP-binding cassette transporter A1/G1(ABCA1/ABCG1).

**Figure 3 biomedicines-12-01832-f003:**
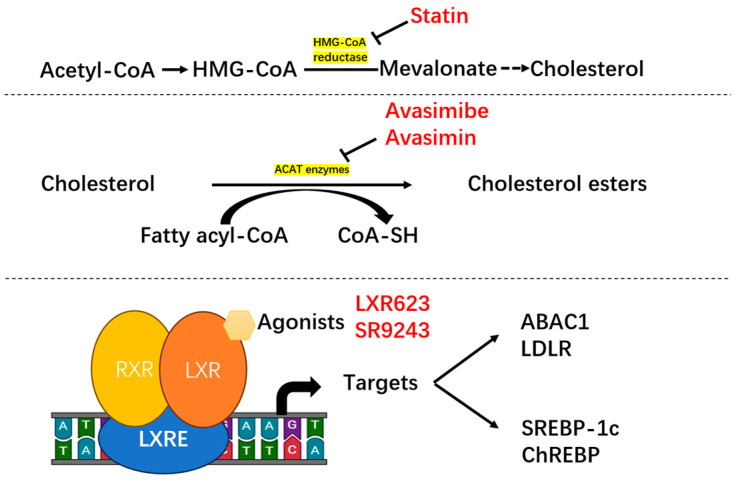
Mechanism for cholesterol metabolism-regulating drugs. Statins can inhibit HMG-CoA reductase to regulate cholesterol metabolism. Avasimibe and Avasimin can regulate cholesterol metabolism by inhibiting the Acetyl-CoA Acetyltransferase (ACAT) enzyme. The liver X receptor (LXR) agonist LXR623 and SR9243 downregulated the expression of the lipogenic gene. (RXR: retinoid X receptor; LXRE: liver X receptor response element; ABCA1: ATP-binding cassette transporter A1; LDLR: low-density lipoprotein receptor; SREBP-1c: sterol regulatory element-binding protein 1c; ChREBP: carbohydrate-responsive element-binding protein).

**Table 1 biomedicines-12-01832-t001:** Anti-cancer therapies that target cholesterol metabolism.

	Reagent	Target	Mechanism	Cancer Type	References
Targeting cholesterol biosynthesis	Statins	HMGCR	Decreased cancer mortality and longer survival, according to retrospective clinical analysis	Prostate cancer	[98]
Targeting cholesterol esterification	Avasimibe (15 mg/kg, PO)	ACAT1	Impaired Wnt–β-catenin signalling through decreased Wnt3a secretion; decreased metastasis; according to preclinical animal models with in vitro analysis (non-obese diabetic/severe combined immunodeficiency mice)	PC-3M prostate cancer model	[100]
	Avasimin (10 mg/kg, IV)		Increased cell apoptosis with no clear cytotoxicity; according to preclinical animal models (BALB/c mice)	PC3 prostate and HCT116 CRC models	[101]
Targeting LXR signaling	LXR623 (30 mg/kg, PO)	LXR	Decreased cellular cholesterol of cancer cells; according to preclinical animal models (C57BL/6 mice)	Clear cell renal cell carcinoma model	[102]
	SR9243 (30 mg/kg, PO)		Repression of lipogenesis and glycolysis of cancer cells; induction of cell apoptosis; according to preclinical animal models (C57BL/6 mice)	Prostate cancer	[102]

Acetyl-CoA Acetyltransferase (ACAT); liver X receptor (LXR); 3-hydroxy-3-methylglutaryl-coenzyme A reductase (HMGCR).

## Data Availability

All data presented in this study are included within the paper.

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
