# Peer review of "Cholesterol Metabolism and Urinary System Tumors"

_biomedicines, 2024, doi:10.3390/biomedicines12081832_

Round 1
Reviewer 1 Report
Comments and Suggestions for Authors
Biomedicines_3135680
Cholesterol metabolism and urinary system tumors
This paper reviewed the correlation of cholesterol metabolism and urinary system tumors. They reviewed research showing that cholesterols or byproduct of cholesterol were involved in tumorigenesis and tumor growth of bladder, prostate and kidney cancer. In page 4, last paragraph should be more developed. We suggest that a better figure diagram should be drawn to show how cholesterols or its by products are involved in different cancer pathway rather than a diagram showing cholesterol metabolism. While the review rather shows the cholesterol metabolism is involved in urinary cancer, about half of the conclusion section is that “cholesterol levels should not be blindly reduced”. The transition after that part to the development of anti-cancer drugs and the advantage of cholesterol targeting drugs is rather contradictory. Also, small typo: in “Abstract:Cancers”, Cancers should not be bold and “Pose a substantial” to “Pose substantial”. While reviews on the correlation between cholesterol and cancer has been already published, we appreciate the fact that relatively new discovery has been added to this review. Small suggestion will be to address that this is not the first review, but state what was updated or newly found in this review.
Major comments:
1. The mechanisms of SREBP, Insig-1, and SCAP mentioned in "2. Normal Cholesterol Metabolism" are not illustrated separately. It would be easier to understand if they were integrated into Figure 1. or shown separately.
2. In 3.2, Prostate Cancer, "metastatic castration-resistant prostate cancer (mCRPC) treatment has undergone tremendous changes with the introduction of novel treatment strategies" ---> It would be good to add a brief explanation of what the new treatment strategies are and how they have changed.
3. At the end of 3.2, Prostate Cancer section, "Cholesterol levels are associated with the occurrence and development of prostate cancer, and this association has racial differences." ---> It would be helpful to briefly mention what the differences are.
4. At the end of 3.3, Kidney Cancer, "Cytochrome P450 Family 27 Subfamily A Member 1 exerts a tumor-suppressive effect on RCC." The sentence is unsourced.
5. The font of the 100th cited sentence in the 4. Targeting cholesterol metabolism for cancer therapy section is different from the other sentences.
6. The Conclusion section seems to lack explanation. ---> "Cholesterol and its metabolic products have unique advantages in adjuvant medication and targeting drug-resistant tumor cells." ---> What is the "unique advantages"? A brief explanation would make the conclusion more convincing.
7. In Figure 2, "Stain" should be changed to "Statin" and the sentence describing it should be Statin, not Stain. Furthermore, next sentence, "Avasimibe and Avasimin can regulate cholesterol metabolism by inhibiting the ACAT enzyme." and the Table 1 font is different from the whole text.
8. Remove redundant content to enhance the focus and readability of the paper. For example, the first sentences of Abstract and Introduction overlap.
9. Some sections are not logically connected. Reorganize the structure of the paper to ensure each section is logically connected.
10. In the part of 4, A comprehensive discussion of various treatment strategies is needed. It is necessary to reinforce the content.
11. The conclusion is somewhat weak. Strengthen the conclusion to clearly highlight the importance of the research and suggest future research directions.
Minor comments:
1. In the part of 4, references should be updated.
2. In page 6, “Avasimin can inhibit PC3 prostate cancer by In-creasing cell apoptosis with no clear cytotoxicity”, where reference 100 is referenced, the font is different.
3. The case of (Fig 1) on page 3 and (fig 2) on page 6 should be unified.
4. In page 4, for example, [44] [39] and [50, 51] should be unified in one form.
5. In page 5, [66 67 68] should be unified to [66-68].
6. In page 5, Abca1 should be unified to ABCA1
7. In paragraph 4. Targeting cholesterol metabolism for cancer therapy, it seems better to add more clinical or in vivo references.
8. In reference, it needs to be modified from reference to references.
9. In reference, the start of the line is different from 1-48 to 49-102. There is one more parenthesis added to 3. It needs to be changed the spacing on 23.
10. In Figure 1. the full names of NPC1 and ABCA1/ABCG1 should be written in figure legend.
11. There are several abbreviations in Table 1. and Figure 2. that do not have full names.
Author Response
Reviewer #1
This paper reviewed the correlation of cholesterol metabolism and urinary system tumors. They reviewed research showing that cholesterols or byproduct of cholesterol were involved in tumorigenesis and tumor growth of bladder, prostate and kidney cancer. In page 4, last paragraph should be more developed. We suggest that a better figure diagram should be drawn to show how cholesterols or its by products are involved in different cancer pathway rather than a diagram showing cholesterol metabolism. While the review rather shows the cholesterol metabolism is involved in urinary cancer, about half of the conclusion section is that “cholesterol levels should not be blindly reduced”. The transition after that part to the development of anti-cancer drugs and the advantage of cholesterol targeting drugs is rather contradictory. Also, small typo: in “Abstract:Cancers”, Cancers should not be bold and “Pose a substantial” to “Pose substantial”. While reviews on the correlation between cholesterol and cancer has been already published, we appreciate the fact that relatively new discovery has been added to this review. Small suggestion will be to address that this is not the first review, but state what was updated or newly found in this review.
Response: We thank the Reviewer for the positive comments regarding the manuscript. We are grateful for the feedback provided and have worked diligently to address all of the concerns. In the following, we respond point-by-point to the Reviewer’s comments. We believe, and hope the Reviewer agrees, that the paper is much improved in content and clarity.
Major comments:
1.The mechanisms of SREBP, Insig-1, and SCAP mentioned in "2. Normal Cholesterol Metabolism" are not illustrated separately. It would be easier to understand if they were integrated into Figure 1. or shown separately.
Response: We appreciate the reviewer’s suggestion. The mechanisms of SREBP, Insig-1, and SCAP shown separately in Fig 1.
Fig 1. The sterol regulatory element-binding protein pathway.
In the presence of cholesterol and oxycholesterols (25-hydroxycholesterol and 27-hydroxycholesterol), the sterol regulatory element-binding protein 2 (SREBP2)–SREBP cleavage-activating protein (SCAP) complex is retained in the endoplasmic reticulum together with insulin-induced gene proteins (INSIGs). In the absence of sterols, INSIGs become ubiquitylated (Ub) and are rapidly degraded. After transport to the Golgi, two proteolytic cleavage enzymes, site-1 protease (S1P) and S2P, release the N-terminal domain of SREBP2. Proteolysis of SREBP1 is not strongly sterol-regulated, but rather is inhibited by polyunsaturated fatty acids (PUFAs) and induced by insulin or high-glucose conditions. SREBP1 activation remains incompletely understood. GP78, E3 ubiquitin-protein ligase AMFR.
2.In 3.2, Prostate Cancer, "metastatic castration-resistant prostate cancer (mCRPC) treatment has undergone tremendous changes with the introduction of novel treatment strategies" ---> It would be good to add a brief explanation of what the new treatment strategies are and how they have changed.
Response: We appreciate the reviewer’s suggestion. the new treatment strategies are immunotherapy, poly (ADP-ribose) polymerase (PARP) inhibitor (PARPi) and androgen receptor signalling inhibitor (ARSI), Abiraterone( a specific cytochrome 17α-hydroxylase inhibitor that is critical for androgen synthesis), Enzalutamide(a potent nonsteroidal AR inhibitor that binds to AR with high affinity).
3.At the end of 3.2, Prostate Cancer section, "Cholesterol levels are associated with the occurrence and development of prostate cancer, and this association has racial differences." ---> It would be helpful to briefly mention what the differences are.
Response: We appreciate the reviewer’s suggestion. Total cholesterol was associated with higher fatal prostate cancer risk in White men only. ApoA was associated with higher fatal prostate cancer risk overall, but ApoB was associated with higher fatal prostate cancer risk in Black men only. Please refer to the details for details: (https://www.hopkinsmedicine.org/news/articles/2021/11/cholesterol-prostate-cancer-and-race)
4.At the end of 3.3, Kidney Cancer, "Cytochrome P450 Family 27 Subfamily A Member 1 exerts a tumor-suppressive effect on RCC." The sentence is unsourced.
Response: We appreciate the reviewer’s suggestion. At the end of 3.3, Kidney Cancer, "Cytochrome P450 Family 27 Subfamily A Member 1 exerts a tumor-suppressive effect on RCC." The sentence is sourced from Androutsopoulos VP, Tsatsakis AM, Spandidos DA. Cytochrome P450 CYP1A1: wider roles in cancer progression and prevention. BMC Cancer. 2009;9:187. Published 2009 Jun 16. doi:10.1186/1471-2407-9-187.
5.The font of the 100th cited sentence in the 4. Targeting cholesterol metabolism for cancer therapy section is different from the other sentences.
Response: We sincerely thank the reviewer for careful reading. We have carefully checked the manuscript and corrected this error.
6.The Conclusion section seems to lack explanation. ---> "Cholesterol and its metabolic products have unique advantages in adjuvant medication and targeting drug-resistant tumor cells." ---> What is the "unique advantages"? A brief explanation would make the conclusion more convincing.
Response: We appreciate the reviewer’s suggestion. we thought that It could provide a new avenue for the treatment of drug-resistant urological tumors.
7.In Figure 2, "Stain" should be changed to "Statin" and the sentence describing it should be Statin, not Stain. Furthermore, next sentence, "Avasimibe and Avasimin can regulate cholesterol metabolism by inhibiting the ACAT enzyme." and the Table 1 font is different from the whole text.
Response: We sincerely thank the reviewer for careful reading. We have carefully checked the manuscript and corrected this error.
8.Remove redundant content to enhance the focus and readability of the paper. For example, the first sentences of Abstract and Introduction overlap.
Response: We sincerely thank the reviewer for careful reading. We have carefully checked the manuscript and the repetitions have been corrected.
9.Some sections are not logically connected. Reorganize the structure of the paper to ensure each section is logically connected.
Response: We appreciate the reviewer’s suggestion. We have carefully checked the manuscript and the problems have been corrected.
- In the part of 4, A comprehensive discussion of various treatment strategies is needed. It is necessary to reinforce the content.
Response: We sincerely thank the reviewer for careful reading. This article focuses on the cholesterol metabolism process and the role of cholesterol metabolism in the occurrence and development of urinary system tumors. The clinical application of targeting cholesterol for the treatment of urinary system tumors requires further basic and clinical research. Therefore, a comprehensive discussion was not conducted, and only some existing therapeutic targets were described.
- The conclusion is somewhat weak. Strengthen the conclusion to clearly highlight the importance of the research and suggest future research directions.
Response: We appreciate the reviewer’s suggestion. We have highlighten the importance of the research and suggest future research directions.
Minor comments:
1.In the part of 4, references should be updated.
Response: We appreciate the reviewer’s suggestion. references have updated.
2.In page 6, “Avasimin can inhibit PC3 prostate cancer by In-creasing cell apoptosis with no clear cytotoxicity”, where reference 100 is referenced, the font is different.
Response: We sincerely thank the reviewer for careful reading. We have carefully checked the manuscript and corrected this error.
3.The case of (Fig 1) on page 3 and (fig 2) on page 6 should be unified.
Response: We sincerely thank the reviewer for careful reading. We have carefully checked the manuscript and corrected this error.
4.In page 4, for example, [44] [39] and [50, 51] should be unified in one form.
Response: We sincerely thank the reviewer for careful reading. We have carefully checked the manuscript and corrected this error.
5.In page 5, [66 67 68] should be unified to [66-68].
Response: We sincerely thank the reviewer for careful reading. We have carefully checked the manuscript and corrected this error.
6.In page 5, Abca1 should be unified to ABCA1
Response: We sincerely thank the reviewer for careful reading. We have carefully checked the manuscript and corrected this error.
7.In paragraph 4. Targeting cholesterol metabolism for cancer therapy, it seems better to add more clinical or in vivo references.
Response: We appreciate the reviewer’s suggestion. but due to the limited clinical investigations and basic experiments targeting cholesterol metabolism, it is currently not.
8.In reference, it needs to be modified from reference to references.
Response: We sincerely thank the reviewer for careful reading. We have carefully checked the manuscript and corrected this error.
9.In reference, the start of the line is different from 1-48 to 49-102. There is one more parenthesis added to 3. It needs to be changed the spacing on 23.
Response: We sincerely thank the reviewer for careful reading. We have carefully checked the manuscript and corrected this error.
10.In Figure 1. the full names of NPC1 and ABCA1/ABCG1 should be written in figure legend.
Response: We sincerely thank the reviewer for careful reading. We have carefully checked the manuscript and corrected this error. Niemann-Pick disease type C1/2(NPC2/NPC1), ATP-binding cassette transporter A1/G1(ABCA1/ABCG1).
- There are several abbreviations in Table 1. and Figure 2. that do not have full names.
Response: We appreciate the reviewer’s suggestion. We have completed the full names of the above abbreviations. (RXR: Retinoid X Receptor. LXRE: Liver X Receptor Response Element. ABCA1: ATP-binding Cassette Transporter A1. LDLR: Low-Density Lipoprotein Receptor. SREBP-1c: Sterol Regulatory Element-Binding Protein 1c. ChREBP: Carbohydrate-Responsive Element-Binding Protein)

Reviewer 2 Report
Comments and Suggestions for Authors
In the review entitled “Cholesterol metabolism and urinary system tumors”, the authors provided a literature review and analysis of the existing research on tumors of the urinary system, notably, renal, bladder, and prostate cancer. Prostate cancer is the most common malignant tumor in men, second only to lung cancer, while bladder cancer is more prevalent in men. The development of tumors is influenced by factors such as smoking, obesity, and genetics. As cholesterol metabolism plays an important role in tumor progression, further research is needed to develop novel methods of prevention, diagnosis, and treatment. The authors provided one interesting table summarizing the anti-cancer therapies that target cholesterol metabolism. Additionally, the authors provided two interning figures summarizing the homeostasis of intracellular cholesterol metabolism and the mechanism for cholesterol metabolism-regulating drugs. The summarized information is interesting and the review is clearly written.
Comments:
1) The provided sections read like narration for the evidence of discussed points without critical aspects/reflection points. At the end of each section, a take-home message is advised to be provided.
2) To enhance the insights derived from Table 1, the authors should specify:
i) type of study whether this data is derived from a clinical trial, animal study (add whether mouse, rat, etc), or cell line.
ii) the dose and route of administration used.
iii) adverse events (if any), and retention in trial.
3) The authors are advised to make the table/figure captions stand-alone. To this end, authors are advised to provide a list of abbreviations describing the full names of all the listed abbreviations in the table/figure.
4) To avoid readers’ confusion, the authors are advised to clearly describe in the narration of previous literature whether these data are derived from clinical studies or from experimental studies. This point needs to be carefully addressed by the authors in the entire manuscript.
5) The work lacks future directions that include limitations and what is the next step to translate these findings to clinical settings.
6) The term “conclusion” section should be replaced by “Conclusions and future directions”.
7) In p. 6, the authors state “Larsen demonstrated that….[98]”, Lee reported that ……【99].
The authors are advised to re-write the statement as “Larsen et al. [98] demonstrated that ….”. Lee and co-workers reported that ……【99]. Please, address this point in the entire manuscript.
8) The in-text citation style needs to be reformatted according to the instructions of the journal. For example, in the first paragraph on page 1, the authors state” Compared with 1990, the global incidence of KC, UBC, and PCa has increased by 155%, 123%, and 169%, respectively【1】【6】”. This should be corrected as [1,6].
9) Figures and tables should be placed immediately after their corresponding descriptions in the text.
Author Response
Reviewer #2
In the review entitled “Cholesterol metabolism and urinary system tumors”, the authors provided a literature review and analysis of the existing research on tumors of the urinary system, notably, renal, bladder, and prostate cancer. Prostate cancer is the most common malignant tumor in men, second only to lung cancer, while bladder cancer is more prevalent in men. The development of tumors is influenced by factors such as smoking, obesity, and genetics. As cholesterol metabolism plays an important role in tumor progression, further research is needed to develop novel methods of prevention, diagnosis, and treatment. The authors provided one interesting table summarizing the anti-cancer therapies that target cholesterol metabolism. Additionally, the authors provided two interning figures summarizing the homeostasis of intracellular cholesterol metabolism and the mechanism for cholesterol metabolism-regulating drugs. The summarized information is interesting and the review is clearly written.
Response: We thank the Reviewer for the positive comments regarding the manuscript. We are grateful for the feedback provided and have worked diligently to address all of the concerns. In the following, we respond point-by-point to the Reviewer’s comments. We believe, and hope the Reviewer agrees, that the paper is much improved in content and clarity.
Comments:
1) The provided sections read like narration for the evidence of discussed points without critical aspects/reflection points. At the end of each section, a take-home message is advised to be provided.
Response: We appreciate the reviewer’s suggestion. We have provided some take-home message at the end of some sections. For example, in 5. Conclusions and future directions, we added that Cholesterol and its metabolic products have unique advantages in adjuvant medication and targeting drug-resistant tumor cells, It could provide a new avenue for the treatment of drug-resistant urological tumors. Current treatment strategies such as targeting cholesterol synthesis, cholesterol esterification, liver X receptor (LXR) signaling, and combination strategies have shown great promise for treating urinary system tumors. However, in the future, interventions targeting other cholesterol metabolism points can still be explored. Cholesterol and its metabolic products are central to the treatment of urinary system tumors in the future.
2) To enhance the insights derived from Table 1, the authors should specify:
- i) type of study whether this data is derived from a clinical trial, animal study (add whether mouse, rat, etc), or cell line.
- ii) the dose and route of administration used.
iii) adverse events (if any), and retention in trial.
Response: We appreciate the reviewer’s suggestion. We have specified the type of study and the dose and route of administration used in the table 1. But those studies did not report significant adverse events and retention in trial.
Table 1. Anti-cancer therapies that target cholesterol metabolism.
|
Reagent |
Target |
Mechanism |
Cancer type |
References |
|
|
Targeting cholesterol biosynthesis |
Statins |
HMGCR |
Decreased cancer mortality and longer survival, according to retrospective clinical analysis |
prostate cancer |
[99] |
|
Targeting cholesterol esterification |
Avasimibe (15 mg/kg, PO) |
ACAT1 |
Impaired Wnt–β-catenin signalling through decreased Wnt3a secretion; decreased metastasis. according to preclinical animal models with in vitro analysis (Non-Obese Diabetic/Severe Combined Immunodeficiency mice) |
PC-3M prostate cancer model |
[101] |
|
Avasimin (10 mg/kg, IV) |
Increased cell apoptosis with no clear cytotoxicity. according to preclinical animal models (BALB/c mice) |
PC3 prostate and HCT116 CRC models |
[102] |
||
|
Targeting LXR signalling |
LXR623 (30 mg/kg, PO) |
LXR |
Decreased cellular cholesterol of cancer cells according to preclinical animal models (C57BL/6 mice) |
Clear cell renal cell carcinoma model |
[103] |
|
SR9243 (30 mg/kg, PO) |
Repression of lipogenesis and glycolysis of cancer cells; induction of cell apoptosis according to preclinical animal models (C57BL/6 mice) |
prostate cancer |
[103] |
Acetyl-CoA Acetyltransferase (ACAT), Liver X Receptor (LXR), 3-hydroxy-3-methylglutaryl-coenzyme A reductase(HMGCR).
3) The authors are advised to make the table/figure captions stand-alone. To this end, authors are advised to provide a list of abbreviations describing the full names of all the listed abbreviations in the table/figure.
Response: We appreciate the reviewer’s suggestion. We have completed the full names of the above abbreviations above each table or figure. (RXR: Retinoid X Receptor. LXRE: Liver X Receptor Response Element. ABCA1: ATP-binding Cassette Transporter A1. LDLR: Low-Density Lipoprotein Receptor. SREBP-1c: Sterol Regulatory Element-Binding Protein 1c. ChREBP: Carbohydrate-Responsive Element-Binding Protein)
4) To avoid readers’ confusion, the authors are advised to clearly describe in the narration of previous literature whether these data are derived from clinical studies or from experimental studies. This point needs to be carefully addressed by the authors in the entire manuscript.
Response: We appreciate the reviewer’s suggestion. We have described those previous literatures whether these data are derived from clinical studies or from experimental studies. For example, in part 4, Impeding active cholesterol metabolism, for example, by inhibiting the mevalonate pathway, is a feasible anti-tumor strategy (Table 1). This can be attributed to the functions of cholesterol metabolism in cancer progression. The key therapeutic targets are as follows: targeting cholesterol synthesis, cholesterol esterification, liver X receptor (LXR) signaling, and combination strategies. For example, Larsen et al. (clinical research)[99] demonstrated that statin use is associated with reduced PCa-related mortality. Lee and co-workers reported that inhibited cholesterol esterification suppresses PCa metastasis by impairing the wingless-related integration Wnt/β-catenin pathway. Thus, they used Avasimibe to target ACAT1 and suppress tumors (experimental research)[100,101]. Future more, Lee and co-workers found that Avasimin can inhibit PC3 prostate cancer by Increasing cell apoptosis with no clear cytotoxicity (experimental research)[102]. Wu and co-workers targeted the transcription factor receptor LXR to treat ccRCC (experimental research)[103]. Enzalutamide combined with simvastatin can inhibit PCa by restricting surface receptor signaling, thus sensitizing the blockade of human epidermal growth factor receptor 2 or androgen-receptor[104](Fig 3). While several clinical and basic studies have found that drugs targeting cholesterol metabolism can inhibit the progression of genitourinary tumors, evidence remains limited. Future research is needed for further validation, as well as exploration of other drugs targeting cholesterol metabolism.
5) The work lacks future directions that include limitations and what is the next step to translate these findings to clinical settings.
Response: We appreciate the reviewer’s suggestion. We have added the next step to translate these findings to clinical settings, For example, in 5. Conclusions and future directions, we added that Cholesterol and its metabolic products have unique advantages in adjuvant medication and targeting drug-resistant tumor cells, It could provide a new avenue for the treatment of drug-resistant urological tumors. Current treatment strategies such as targeting cholesterol synthesis, cholesterol esterification, liver X receptor (LXR) signaling, and combination strategies have shown great promise for treating urinary system tumors. However, in the future, interventions targeting other cholesterol metabolism points can still be explored. Cholesterol and its metabolic products are central to the treatment of urinary system tumors in the future.
6) The term “conclusion” section should be replaced by “Conclusions and future directions”.
Response: We appreciate the reviewer’s suggestion. We have replaced the term “conclusion” section by “Conclusions and future directions”.
7) In p. 6, the authors state “Larsen demonstrated that….[98]”, Lee reported that ……【99].The authors are advised to re-write the statement as “Larsen et,al.[98]demonstrated that ….”. Lee and co-workers reported that ……【99]. Pleas-e, address this point in the entire manuscript.
Response: We sincerely thank the reviewer for careful reading. We have carefully checked the manuscript and corrected this error.
8) The in-text citation style needs to be reformatted according to the instructions of the journal. For example, in the first paragraph on page 1, the authors state” Compared with 1990, the global incidence of KC, UBC, and PCa has increased by 155%, 123%, and 169%, respectively【1】【6】”. This should be corrected as [1,6].
Response: We sincerely thank the reviewer for careful reading. We have carefully checked the manuscript and corrected this error.
9) Figures and tables should be placed immediately after their corresponding descriptions in the text.
Response: We sincerely thank the reviewer for careful reading. We have placed Figures and tables should be placed immediately after their corresponding descriptions in the text.

Reviewer 3 Report
Comments and Suggestions for Authors
Dear Auther
The study titled "Cholesterol Metabolism and Urinary System Tumors" provides a comprehensive exploration of the relationship between cholesterol metabolism and tumors in the urinary system, focusing on kidney, bladder, and prostate cancers. The paper's strengths include its detailed data on how cholesterol influences tumor growth and metastasis, highlighting various mechanisms by which cholesterol affects cancer cell behavior. Additionally, the study offers valuable insights into therapeutic strategies targeting cholesterol metabolism. However, the paper is too lengthy and can be condensed for better readability, with some sections being overly technical and potentially inaccessible to all readers. There is also a lack of recent clinical trial data to support some of the claims made. The introduction is clear but could benefit from a more concise summary of key points, and while the section on cholesterol's role in cancer cell signaling is well-detailed, it could be more concise. The discussion on therapeutic strategies is informative but needs more current data. Overall, the study is well-researched and presents important findings but would benefit from being more concise and including more recent clinical data. Therefore, a minor revision is recommended
Minor editing of English language required
Author Response
We thank the Reviewer for the positive comments regarding the manuscript. We are grateful for the feedback provided and have worked diligently to address all of the concerns.

Round 2
Reviewer 1 Report
Comments and Suggestions for Authors
The authors have described reviewer comments in detail
Reviewer 2 Report
Comments and Suggestions for Authors
The authors have adequately addressed the raised comments.